# Mechanistic Insight from Preclinical Models of Parkinson’s Disease Could Help Redirect Clinical Trial Efforts in GDNF Therapy

**DOI:** 10.3390/ijms222111702

**Published:** 2021-10-28

**Authors:** Karen M. Delgado-Minjares, Daniel Martinez-Fong, Irma A. Martínez-Dávila, Cecilia Bañuelos, M. E. Gutierrez-Castillo, Víctor Manuel Blanco-Alvarez, Maria-del-Carmen Cardenas-Aguayo, José Luna-Muñoz, Mar Pacheco-Herrero, Luis O. Soto-Rojas

**Affiliations:** 1Centro de Investigación y de Estudios Avanzados del Instituto Politécnico Nacional, Departamento de Fisiología, Biofísica y Neurociencias, Av. Instituto Politécnico Nacional No. 2508, Ciudad de México 07360, Mexico; km95.delgado@gmail.com (K.M.D.-M.); daniel.martinezfong@cinvestav.mx (D.M.-F.); irmaalice@yahoo.com (I.A.M.-D.); 2Programa de Doctorado en Nanociencias y Nanotecnología, Centro de Investigación y de Estudios Avanzados del Instituto Politécnico Nacional, Av. Instituto Politécnico Nacional No. 2508, Ciudad de México 07360, Mexico; 3Nanoparticle Therapy Institute 404 Avenida Monte Blanco Aguascalientes, Aguascalientes 20120, Mexico; 4Coordinación General de Programas de Posgrado Multidisciplinarios, Programa de Doctorado Transdisciplinario en Desarrollo Científico y Tecnológico para la Sociedad, Centro de Investigación y de Estudios Avanzados del Instituto Politécnico Nacional, Av. Instituto Politécnico Nacional No. 2508, Ciudad de México 07360, Mexico; cebanuelos@cinvestav.mx; 5Centro Interdisciplinario de Investigaciones y Estudios sobre Medio Ambiente y Desarrollo, Departamento de Biociencias e Ingeniería, Instituto Politécnico Nacional, 30 de Junio de 1520 s/n, Ciudad de México 07340, Mexico; ainegue9@gmail.com; 6Facultad de Medicina, Benemérita Universidad Autónoma de Puebla, 13 Sur 2702, Puebla 72420, Mexico; victor.blancoa@correo.buap.mx; 7Facultad de Enfermería, Benemérita Universidad Autónoma de Puebla, Av. 25 Poniente 1304, Los Volcanes, Puebla 72410, Mexico; 8Laboratory of Cellular Reprogramming, Departamento de Fisiología, Facultad de Medicina, UNAM. Av. Universidad No. 3000, Ciudad de México 04510, Mexico; mcardenasaguayo@unam.mx; 9National Dementia BioBank, Ciencias Biológicas, Facultad de Estudios Superiores Cuautitlán, Universidad Nacional Autónoma de México, México City 53150, Mexico; jluna_tau67@comunidad.unam.mx; 10Banco Nacional de Cerebros, Universidad Nacional Pedro Henríquez Ureña, Santo Domingo 1423, Dominican Republic; 11Neuroscience Research Laboratory, Faculty of Health Sciences, Pontificia Universidad Católica Madre y Maestra, Santiago de los Caballeros 51000, Dominican Republic; mpacheco@pucmm.edu.do; 12Facultad de Estudios Superiores Iztacala, UNAM, Mexico City 54090, Mexico

**Keywords:** neurodegenerative diseases, α-synuclein, neuroinflammation, neurotrophic factors, anti-α-synuclein therapy, anti-inflammatory therapy, neurorestoration

## Abstract

Parkinson’s disease (PD) is characterized by four pathognomonic hallmarks: (1) motor and non-motor deficits; (2) neuroinflammation and oxidative stress; (3) pathological aggregates of the α-synuclein (α-syn) protein; (4) neurodegeneration of the nigrostriatal system. Recent evidence sustains that the aggregation of pathological α-syn occurs in the early stages of the disease, becoming the first trigger of neuroinflammation and subsequent neurodegeneration. Thus, a therapeutic line aims at striking back α-synucleinopathy and neuroinflammation to impede neurodegeneration. Another therapeutic line is restoring the compromised dopaminergic system using neurotrophic factors, particularly the glial cell-derived neurotrophic factor (GDNF). Preclinical studies with GDNF have provided encouraging results but often lack evaluation of anti-α-syn and anti-inflammatory effects. In contrast, clinical trials have yielded imprecise results and have reported the emergence of severe side effects. Here, we analyze the discrepancy between preclinical and clinical outcomes, review the mechanisms of the aggregation of pathological α-syn, including neuroinflammation, and evaluate the neurorestorative properties of GDNF, emphasizing its anti-α-syn and anti-inflammatory effects in preclinical and clinical trials.

## 1. Introduction

Parkinson’s disease (PD) is the second most common neurodegenerative disorder that affects around 10 million people worldwide. Furthermore, prospective studies predict that the PD prevalence will continue increasing for the next 30 years caused by the expanding life expectancy of the general population [1,2,3]. PD is well known through the disabling motor deficits, including bradykinesia, tremor, stiffness, and postural instability, which arise from the progressive loss of dopaminergic (DA) neurons in the *substantia nigra pars compacta* (SNpc) [1,4]. Besides, non-motor alterations, such as olfactory loss and mood changes, also occur in PD and can appear years or decades before the motor deficit manifestation [5].

Another pathological hallmark of PD is the presence of Lewy bodies and Lewy neurites, mainly composed of misfolded α-synuclein (α-syn) [6,7]. This protein plays different physiological roles in neuronal homeostasis, predominantly in clustering and releasing synaptic vesicles during neuronal plasticity [8]. Unfortunately, many events can modify the structure of α-syn converting it into a pathological protein that gives rise and extends the PD process [9]. Experimental evidence suggests that the neurotoxicity of α-syn comes from its transformation into insoluble aggregates, either oligomers or fibrils [10]. Aggregates of α-syn are known to activate pathological intracellular signaling in neurons and glial cells, leading to membrane disruption, mitochondrial dysfunction, homeostatic alterations, and finally, neuronal death [11]. However, the exact mechanism in the DA neurodegeneration remains unknown [9]. Recently, an increasing number of publications associate pathological α-syn with neuroinflammation, the latter unveiled through microglia activation, astrocyte reactivity, inflammatory mediator release, blood-brain barrier (BBB) breakdown, and immune cell infiltration [12,13]. Interestingly, extracellular α-syn aggregates [14] can directly stimulate microglia and astrocytes to produce pro-inflammatory cytokines and initiate neuroinflammation [12,15,16,17]. Moreover, pathological α-syn can also significantly decrease glial cell-derived neurotrophic factor (GDNF) levels and its receptors tyrosine kinase (RET) expression [18,19], thus, eradicating a vital survival pathway of DA neurons in adulthood.

It is still controversial whether the chronic inflammatory environment is the cause or the consequence of neurodegeneration. However, the current consensus is that neuroinflammation can lead to neural degeneration, for which it has become the third pathological hallmark of PD [20,21]. Moreover, the importance of neuroinflammation relies on its ability to launch a vicious circle by recruiting professional immune cells, which release harmful humoral factors that worsen neuroinflammation [22].

Given the lack of regenerative actions of the current pharmacological treatments for PD [23,24], modern therapeutics aim to reduce the neuroinflammation and the pathological α-syn aggregation, both considered causal factors behind the DA neurodegeneration.

Another modern therapeutic strategy uses neurotrophic factors that participate in the differentiation and growth of DA neurons during brain development and promote maintenance and survival in adulthood [1]. GDNF is an attractive candidate that has been tested in PD models in vitro and in vivo because of its potent neuroregenerative action and its apparent anti-α-syn and anti-inflammatory effects [25,26,27,28,29,30,31,32,33,34]. However, the outcomes of clinical trials with GDNF through various administration routes have been disappointing, and some patients have developed severe adverse side effects [35,36]. The uncertainty of GDNF as a therapeutic agent arises from the lack of knowledge about GDNF’s effects on α-syn aggregation and neuroinflammation in preclinical studies and clinical trials. Therefore, the question arises as to whether GDNF is an appropriate therapeutic agent for treating PD or just another dead end of what once seemed to be a promising idea.

In this review, we discuss the newest knowledge on the role of pathological α-syn aggregates in the neuroinflammatory process, two of the pathognomonic hallmarks of PD. In this line, we analyze possible pathophysiological mechanisms through which α-syn could cause neuroinflammation and contribute to the death of DA neurons. Besides, we discuss preclinical and clinical studies on the therapeutic use of GDNF, emphasizing its neuroregenerative action and its anti-α-syn and anti-inflammatory effects.

## 2. The Role of α-Synuclein in Parkinson’s Disease

### 2.1. Structure and Physiology of α-Synuclein

The α-syn, encoded by the *SNCA1/PARK1* gene, is a ubiquitous protein that is abundantly expressed in kidneys and blood cells [37], but highly enriched in the brain, particularly in the presynaptic terminals of the neocortex, hippocampus, *substantia nigra* (SN), thalamus, and cerebellum. Interestingly, it has been found expressed in the cytoplasm of astrocytes and oligodendrocytes in healthy individuals [38,39]. Different scientific approaches have converged in the description of α-syn as an intrinsically disordered protein (IDP), with unusual structural properties [40]. Three distinct domains (Figure 1) confer dynamic structural flexibility and remarkable conformational plasticity [40,41,42,43]; (1) The N-terminal region (amino acids 1–65) confers an α-helical structure involved in lipid membranes binding and possibly promotes α-syn oligomerization; (2) The central region (amino acids 61–95) includes the NACore phosphate-binding loop, which has been implicated in the formation of amyloid fibrils and the stabilization of the pathogenic conformation of α-syn; (3) The C-terminal region (amino-acids 96–140) is associated with the major sites of metal binding and posttranslational modification, involved in modulating the protein structure, physiological functions, and toxicity.

Concerning the physiological roles of this protein, several studies have considered the subcellular localization of α-syn, which ranges from the nucleus to mitochondria and nerve terminals, to propose the following functions [44,45,46,47,48]: (i) neuronal health maintenance; (ii) synaptic plasticity; (iii) membrane biogenesis; iv) mitigation of oxidative stress; (v) regulation of synaptic vesicle trafficking; (vi) neurotransmitter release.

### 2.2. Pathological α-Syn Aggregates and Prion-like Properties

Several lines of evidence have proposed that native α-syn exists predominantly as an IDP monomer, which is typically found in an unfolded state and soluble in the cytosol, minimally phosphorylated in the healthy human brain. However, this dynamic protein can convert to various conformations such as helically folded tetramers resisting aggregation, pathologic oligomers, small aggregates, protofibrils, or irreversible insoluble amyloid fibrils with a stabilizing β-sheet structure [42,49,50]. Most α-syn forms exist in a dynamic equilibrium with each other, but perturbation of neuronal homeostasis is a starting point for pathological α-syn insolubility, self-assembly, β-sheet stacking, and misfolding. Cellular environmental cues combined with genetic factors contribute to the posttranslational modifications of the unfolded monomeric α-syn that lead to dysfunctional, neurotoxic, and pathological aggregates with a high degree of β-sheet structure [51].

Interestingly, all the known mutations associated with familial forms of PD are clustered within the N-terminal region, causing misfolding and/or aggregation of the mutant α-syn [42]. Furthermore, N-terminal acetylation could be critical for both aggregation and function [52]. In the C-terminal region, posttranslational modifications have been described that promote a tendency to protein aggregation. Examples are the C-terminal truncation, which results in increased filament assembly, and the phosphorylation at S129 (pS129), which regulates membrane-binding and enhances interactions with metal ions and other proteins (Figure 1). Highlighting that pS129 α-syn modulates key events in the pathogenesis of synucleinopathy such as: (i) variations in the fibrillar structure; (ii) different propagation properties; (iii) increase in cytotoxicity [53].

Indeed, a diversity of pathogenic properties of the misfolded conformations and accumulating aggregates of α-syn have been associated with: (i) mitochondrial dysfunction; (ii) endoplasmic reticulum stress; (iii) proteostasis dysregulation; (iv) synaptic impairment; (v) cell apoptosis; (vi) neuroinflammation; and (vii) neurodegeneration [11,54,55,56,57,58].

Notably, the pathological α-syn aggregates may spread from one neuron to another, causing Lewy pathology in the whole brain [59]. However, α-synuclein-positive inclusions have also been found in the cytoplasm of oligodendrocytes, an event that occurs in the α-synucleinopathy called multiple system atrophy. Specifically, Braak et al. (2002) suggested that pathological forms of the α-syn act in a prion-like manner, trafficking between cells in a non-random way/form [59]. They hypothesized that PD pathology initiates in the peripheral nervous system, gaining access to the central nervous system (CNS) through retrograde transport via the olfactory tract and the vagal nerve [59,60]. It has been argued that the release and propagation mechanisms of α-syn between neuroanatomically connected regions can be through exosomes, classical exocytosis, trans-synaptic junctions, tunneling nanotubes, and direct penetration [61]. Last but not least, recent studies have suggested that α-syn misfolding and aggregation trigger microglial activation, leading to neuroinflammation and cellular metabolic stress, enhancing the aggregation and spreading of α-syn and affecting its prion-like transmissibility and pathogenicity (Braak’s hypothesis) [62,63,64,65].

## 3. The Misfolding of α-Synuclein and Its Association with Neuroinflammation in Parkinson’s Disease

A hypothesis claims that chronic neuroinflammation can lead to neuronal damage, neuronal circuitry disturbances, and ultimately, neurodegeneration in PD [20,66]. Hence, chronic neuroinflammation is relevant when considering the pathophysiological mechanisms involved in PD progression and proposing appropriate therapeutic approaches. The brain is an organ susceptible to external stimuli. However, internal stimuli can also alter the delicate homeostasis of the neuronal microenvironment maintained by microglia and astrocytes, considered the brain’s absorptive, excretory, and defense systems [67]. These cells display a Janus-like face because they help eliminate neurotoxins and pathogens, and conversely, they can also cause neuroinflammation, neurotoxicity, and neurodegeneration. Thus, neuroinflammation is a complex pathological condition where different cells and humoral factors converge to resolve the damage as a first intention and latter they aggravate the disease in the long term. The cellular actors are activated microglia, reactive astrocytes, and infiltrated lymphocytes, whereas the humoral factors are a great variety of pro-inflammatory molecules. A resulting critical event from the flare-up between cells and humoral factors activities is the loss of BBB permeability that allows molecules to cross from one side to another of the brain [66].

In PD, microglia activation can arise from several factors or causes. The preference of activated microglia for brain areas enriched with pathological α-syn aggregates supports its close association with the neurodegeneration process in PD [68,69]. Multiple studies have shown that extracellular α-syn stimulates microglial cells to produce pro-inflammatory molecules such as interleukin (IL)-1β, IL-6, tumoral necrosis factor-alpha (TNF-α), cyclooxygenase-2 (COX-2), inducible nitric oxide synthase (iNOS), and reactive oxygen species (ROS) [70,71,72,73,74]. The combined neuroinflammation and oxidative stress can promote the neurodegenerative process and further aggravate it [75,76].

Furthermore, neuron-derived α-syn can stimulate astrocytes to produce and release pro-inflammatory cytokines and chemokines, which in turn can recruit activated microglial cells [12] and differentiate them to an M1 (pro-inflammatory) or M2 (anti-inflammatory) phenotype [77]. Therefore, pathological α-syn also behaves as a chemokine to concentrate activated microglia in the affected anatomical areas in PD [78,79].

The relevance of activated microglia in neuroinflammation and neurodegeneration relies on its ability to convert physiological astrocytes into neurotoxic reactive astrocytes classified as A1 phenotype. Recently, it has been established that IL-1α, TNF, and complement component 1q (C1q) released by activated microglia are sufficient and necessary to detonate the A1 phenotype (Figure 2a) [80]. Furthermore, evidence in vitro and in vivo shows that neurotoxic reactive astrocytes A1 can kill neurons through the secretion of neurotoxins not yet identified [80,81]. Thus, the neurotoxic role of reactive astrocytes A1 has severe implications for PD and other neurodegenerative diseases. It means that neurotoxic astrocytes lost their ability to promote neuronal survival, growth, synaptogenesis, and phagocytosis. Therefore, an effective therapy must also prevent the conversion of neurotoxic reactive astrocytes A1 and block their neurotoxic activity [80,81].

Besides, the interaction between α-syn with glial cells depends on α-syn aggregation state and the receptors responsible for its uptake (Figure 2b). These receptors expressed in glial cells, through which α-syn interact to trigger a neuroinflammatory environment (Table 1 and Figure 2b,c), play a critical role in early PD, considering that the α-syn aggregation process from soluble oligomers to insoluble inclusions occurs in the early phase of the disease [15,82].

On the other hand, an in vitro study demonstrated that monomeric α-syn activates pericytes (critical cells in BBB regulation) to release pro-inflammatory molecules and matrix metalloproteinase-9 (MMP-9) [101]. These findings suggest that pericytes could exacerbate the neuroinflammatory environment and cause the BBB rupture in PD patients (Figure 2c).

Finally, the infiltrated peripheral immune cells, especially CD4+ and CD8+ T lymphocytes, extend and worsen the PD pathology (Figure 2c) [22,102]. This complication shows the critical role BBB breakdown plays in PD, thus becoming a point to control by new therapeutics. Moreover, infiltrated T lymphocytes, recognizing pathological α-syn aggregates as an antigen presented by microglial class II major histocompatibility complex (MHC-II), arise the immune response [103]. The increased infiltration of Th17 cells and reactive Th1 cells differentiated from CD4+ lymphocytes in PD brains proves the immune response’s involvement in this neuropathology [102,104]. Therefore, these antecedents support the immune response as another mechanism of neuronal death and promote the development of effective immunotherapies.

In summary, we propose the following sequential cellular events that lead to neuroinflammation in early PD (Figure 2). First, misfolded α-syn binds to microglial receptors causing differentiation of the microglia into the M1 phenotype. Then, M1 microglia release TNF-α, IL1α, and C1q inducing the conversion of astrocytes into neurotoxic reactive astrocytes A1. Both, the M1 microglia and A1 astrocytes release pro-inflammatory cytokines to open the BBB and chemokines to attract CD4+ and CD8+ cells, thus completing an immune response against misfolded α-syn. Moreover, the action of neurotrophic factors and other protective molecules released by astrocytes A2 is overpassed by the neuroinflammatory events. Altogether, the unknown molecules released by neurotoxic reactive astrocytes A1, the pro-inflammatory cytokines, and the cellular and humoral response of professional immune cells converge to kill the DA neurons in the early stages of PD. Therefore, new antiparkinsonian therapies should prevent α-syn binding to its glial receptors, block microglial M1 and astrocytic A1 activities, strengthen BBB permeability and avoid activation of the immune response. However, present therapeutic effects are insufficient for an integral PD therapy since the structural and functional restoring of the nigrostriatal dopaminergic system is not accomplished. To this purpose, different approaches based on neurotrophic factors have been tested in vitro, in vivo, and clinical trials. GDNF is an attractive therapeutic candidate because of its potent neurotrophic effects on DA neurons during development and adulthood and its potential anti-inflammatory and anti-α-syn therapeutic effects (see below).

## 4. Glial Cell-Derived Neurotrophic Factor in Parkinson’s Disease

### 4.1. Structure, Signaling Pathways, and Function

GDNF is the first neurotrophic factor member of the GDNF family ligands isolated by Lin and coworkers in 1993 from the rat glial cell line B49 [34,105]. GDNF has been proposed as a potent therapeutic agent for PD treatment based on its positive effects on the survival and morphological differentiation of embryonic midbrain DA neurons in culture [34]. GDNF is codified by the *GDNF* gene and translated as a precursor of 211 amino acids, which, after proteolytic processing, is secreted as a mature protein of 134 amino acids [34].

The active form of GDNF is a disulfide-bonded homodimer composed of two β-stranded finger loops and a helical heel that binds to the GDNF family receptor α1 (GFRα1) on its finger domain, forming the high-affinity GDNF-GFRα1 complex [106,107,108]. This complex can activate different signaling pathways depending on the co-receptors to which it binds, such as the receptor tyrosine kinase (RET) or the neural cell adhesion molecule (NCAM) [106,109]. In the first case, the GDNF-GFRα1 complex binds two RET molecules inducing RET dimerization and autophosphorylation of specific tyrosine residues (Y905, Y1015, Y1062, and Y1096). These phosphorylated residues operate as docking sites for signaling adaptor proteins that initiate the signaling of the mitogen-activated protein kinase (MAPK), phosphatidylinositol-3-kinase (PI3K)/AKT, and Src pathways, which finally turn on genetic programs for neural proliferation, differentiation, migration, maintenance, survival, and neurite outgrowth [1,106,110,111]. Instead, the binding of the GDNF-GFRα1 complex with NCAM activates the focal adhesion kinase (FAK)/Fyn pathway (independently of RET signaling) that promotes axonal growth in hippocampal and cortical neurons, as well as Schwann cells migration [109]. Interestingly, multiple GDNF homodimers can simultaneously interact with syndecan-3 (independently of GFRα1 and RET signaling) and activate the Src kinase pathway, resulting in migration of cortical neurons and neurite outgrowth [112].

Under physiological conditions, GDNF is expressed in soft tissue, testis, kidney, adrenal gland, parathyroid gland, placenta, gastrointestinal tract, spinal cord, and multiple brain nuclei [113,114,115]. In the CNS, GDNF expression increases during embryonic development and decreases in adulthood, restricting itself to specific brain areas such as the cortex, hippocampus, striatum (STR), SN, thalamus, cerebellum, and spinal cord [114,116]. GDNF plays a vital role in regulating kidney morphogenesis, enteric system development, and parasympathetic neuron proliferation and migration. Of importance for this review, GDNF is well known for its potent effect on the maturation, maintenance, and survival of DA neurons [34,117,118,119]. Unfortunately, GDNF expression and signaling are dysregulated in neurodegenerative diseases, including PD [19,120]. Thus, suppression of endogenous GDNF due to PD renders it unavailable to contribute to the neural restorative process.

### 4.2. GDNF Alterations in Parkinson’s Disease

PD patients consistently show GDNF protein depletion in the surviving DA neurons of SNpc compared to healthy subjects, thus, partly explaining the lack of restorative effects on the nigrostriatal system [120]. Similarly, in other regions of the nigrostriatal system, the putamen nucleus shows a significant decrease in GDNF protein levels [116]. Interestingly, reduction in GDNF protein levels has also been found in the hippocampus of PD patients, associated with cognitive decline even in the absence of neuronal loss [121]. Hence, GDNF might perform physiological actions in hippocampal neurotransmission similar to those of BDNF [122]. However, in the remaining brain areas such as the cerebellum and frontal cortex, GDNF protein levels do not change [116]. On the other hand, special interest has arisen in the signaling effectors, downstream of GDNF. In the putamen of PD patients, the mRNA expression levels for GFRα1 and RET remain unchanged [123]. Moreover, gene expression analysis in the SN of sporadic PD patients exposed no significant change in RET, which was corroborated in α-syn transgenic mice [124]. Nevertheless, mRNA and gene expression give rise to many speculative mechanisms without a physiological basis since the presence of the protein was not explored.

Indeed, the protein levels for RET receptor and its downstream signaling molecule phosphor-ribosomal protein S6 (p-rpS6) were severely decreased in DA neurons with α-syn inclusions of PD patients. These findings were recapitulated in a non-human primate model overexpressing the human mutant α-syn A53T to validate the human findings. Nevertheless, the remaining RET expression in the nigral neurons is enough to activate the downstream molecule p-rpS6, and the synucleinopathy is not an impediment for GDNF to express its trophic effect [19]. The conflicting results obtained regarding GDNF downstream activation of its signaling molecules could be due to the type of method used to evaluate the presence of these molecules. Whereas no changes are observed in gene expression, assays at the protein level have challenged the earlier conclusions.

### 4.3. Neurotrophic Effects of GDNF in Preclinical Assays

Rodents (Table 2, a) and non-human primates (Table 2, b) with a unilateral or bilateral lesion of the nigrostriatal pathway using specific neurotoxins to model PD have been used to show the preventive and restorative effects of GDNF on DA neuron survival, levels of dopamine, and its catabolites, and motor deficits (see reviews [125,126,127,128,129,130]). However, animal models have significant limitations of applicability to the human disease process [131]. Depending on the selected model, several works have evidenced the relevance of the timing and route of GDNF administration, its concentration or dose, the type of therapy, drug delivery vehicle, and the status of PD progression [35,132,133,134,135,136,137,138]. Specifically, the timing between GDNF administration and exposure to the neurotoxic agent (PD neurotoxin-based models) is critical in achieving optimal protection [139]. In addition, the site of GDNF administration is also critical to achieving functional benefits. For example, a study explored a GDNF in the PD neurotoxin-based model the administration in the STR, SN, or lateral ventricle. The main conclusion was that the preservation of motor functions requires protection of the striatal axon terminals, which can be achieved only by intrastriatal, but not nigral or intraventricular administration of GDNF [140]. Assessment of biochemical, histological, and behavioral changes and a selection of distinct methodologies and scores for determining motor function and behavioral improvements have been used to confirm proper GDNF targeting to the SN, avoiding leakages or ligand-receptor binding in other brain areas.

Lack of clarity on the mechanism of GDNF action could lead to an inappropriate model selection for preclinical therapeutic approaches. Many animal models have been developed to understand PD pathophysiology and propose therapeutic targets. However, not all of them mimic the pathognomonic hallmarks of PD, a requirement for qualifying as a suitable animal PD model [141,142]. Our research group recently developed an animal model by a single intranigral administration of β-sitosterol β-D-glucoside (BSSG) [143,144,145]. This animal PD model is promising because of its following characteristics: (1) development of motor and non-motor alterations; (2) slow and progressive death of DA neurons; (3) spreading and aggregation of pathological α-syn; and (4) neuroinflammation and peripheral immune infiltration. Therefore, it is a suitable animal model reproducing the four pathognomonic hallmarks of PD to test different therapies, including GDNF administration (see below). Furthermore, the outcome of the therapy assays in the stereotaxic BSSG model can predict their efficacy in clinical trials more accurately.

On a final note, the large GDNF homodimer structure impedes the BBB crossing after its systemic administration. Consequently, translational research has focused on direct GDNF protein infusion or vector-mediated GDNF delivery to the brain by several less invasive strategies that enhance vector biodistribution. In this regard, focused ultrasound-assisted delivery, which allows systemic gene delivery, is being used in clinical trials [35,146,147]. Moreover, the perspectives depend on significant improvements to (i) PD preclinical models, (ii) clinical striatal delivery methods, (iii) alternate less invasive methods and (iv) clinical designs that include early PD diagnosis for optimal patient selection. Thus, if GDNF is to be taken forward, every aspect of the clinical design must be optimized. Altogether, the evidence available preserves a cautious optimism for the future of GDNF as a promising target in the treatment of PD progression.

### 4.4. Anti-α-Synuclein Effect of GDNF in Parkinson’s Disease Preclinical Trials

The neuroregenerative features of GDNF have been extensively demonstrated in parkinsonian animals, but its anti-α-syn effect has been evaluated only by five studies (Figure 3) [25,27,170,171,172] whose results are contradictory (Table 3). Besides, four animal models have been used to evaluate the GDNF effect on α-syn. Two models have been developed overexpressing wild-type or A30P mutant α-syn using lentiviral or AAV transduction in the SN or STR. Another is the parkin transgenic mice that raised α-syn levels. The last model is developed by the injection of preformed fibrils (PFFs) of human α-syn in the STR, which has the advantage of triggering pathological α-syn aggregation, and then, the aggregates spread all over the brain, causing α-synucleinopathy where seeded.

A line of evidence shows that GDNF therapy administrated before or after α-syn induction fails to prevent or reduce α-syn levels [170,171,172]. The GDNF failure has been speculated to depend on the stage of the neurodegeneration, being effective at the early stage, when there are sufficient surviving neurons to respond to the therapeutic effect [173]. Another explanation for the lack of GDNF neuroprotective and anti-α-syn effects in some experiments could be related to the PD model employed. Thus, there is no convincing evidence that the high levels of wild or A30P mutant α-syn in viral vector transductions can trigger all features of α-synucleinopathy, i.e., significant neurodegeneration of the nigrostriatal system, motor disabilities, and spreading of pathological α-syn aggregates [171]. This approach needs complementation with an injection of PFF to detonate α-synucleinopathy. Since the discussed studies lack this complementary approach, it is difficult to accept the outcome of GDNF failure as conclusive. Another hypothesis is that GDNF failure can be accounted for by the disruption in the GDNF/RET/Nurr1 signaling pathway caused by pathological α-syn aggregates [172]. However, findings in human brain samples of PD patients refute this hypothesis [19].

On the contrary, an anti-α-syn effect of GDNF was evidenced in Parkin Q311X(A) mice with the injection of GDNF-expressing macrophages [27]. Thus, a dual mechanism of the GDNF beneficial effects in this animal model has been taught to be an anti-inflammatory effect of M2 subtype macrophages and the GDNF/Ret pathway activation [25].

Future research on GDNF should be carried out in the stereotaxic BSSG model of PD, where GDNF effects can be evaluated on neuroinflammation, DA neurodegeneration, α-synucleinopathy, and motor and non-motor deficits.

## 5. Anti-Inflammatory Effects of GDNF in Preclinical Assays

### 5.1. GDNF Anti-Inflammatory Effects in Neurodegenerative Disease Models

In recent years, particular attention has focused on neuroinflammation and its relationship with neurodegenerative disease [174]. A few studies have investigated the anti-inflammatory properties of GDNF in neurodegenerative disease models, such as PD, Alzheimer’s disease (AD), amyotrophic lateral sclerosis (ALS), and multiple sclerosis (MS; Table 4 and Figure 3) [26,27,28,30,31,32,33,175,176,177]. In PD models, GDNF anti-inflammatory effect is mediated by the modulation of microglial activation [27,28,29,30,31,33]. The proposed molecular mechanisms are the activation of the GFRα1-RET complex, the possible inhibition of the FAK pathway, and the reduction in p38 phosphorylation, a component of the p38-MAPK signal transduction pathway [31,33]. In addition, GDNF administration decreases reactive astrocytes, pro-inflammatory and pro-apoptotic mediators, and oxidative stress [26,27,30,31,32]. However, a disadvantage of these PD animal models is that they present acute neuroinflammation, contrary to what was observed in PD [178].

In the AD model, GDNF decreased the microglia-released pro-inflammatory cytokines by reducing the phosphorylation of YAP from the Hippo/YAP pathway [175]. Although the anti-inflammatory effect of GDNF on ALS is not significant, a slight decrease can be seen in the microglial inflammatory marker CD11b, which suggests that GDNF probably acts as an anti-inflammatory molecule [176]. However, different factors, such as an insufficient GDNF concentration or the neuroinflammation severity, could affect GDNF action. Finally, in the MS model, GDNF reduces the amount and size of inflammatory infiltrates in the STR and the extent of NSC astrocytic differentiation, possibly exerting these effects through the activation of myeloid dendritic cells and subsequent restriction of the expansion of T cells [177]. This GDNF effect could also block the peripheral immune cells infiltration to the brain in PD, thus reducing the neuroinflammatory process. We suggest that the decrease in reactive astrocytes could be the result of the reduction in microglial activation, due to the diminish in their release of pro-inflammatory mediators such as IL-1α, TNF-α, and C1q, which were previously mentioned as the factors that trigger the activation of A1 astrocytes (Figure 3). Given that GDNF’s anti-inflammatory properties were only recently described, many questions remain regarding its mechanism of action in different diseases. We hypothesize that the variation in the anti-inflammatory effects of GDNF could be the consequence of the different pathological properties induced by the misfolding of each set of proteins characteristic of each neurodegenerative disease. Nevertheless, the new approach using GDNF to prevent neuroinflammation at early steps in neurodegeneration could be the light at the end of the road to treat some types of neurodegenerative diseases.

### 5.2. GDNF Anti-Inflammatory Effect in Other Neurological Disease Models

GDNF therapy has been similarly used in other neurological diseases where negative regulation of GDNF precedes nerve damage. The rationale for the use of GDNF is based on the following properties: (i) GDNF promotes the regeneration of sensory axons; (ii) GDNF attenuates neuropathic pain; (iii) GDNF reduces the volume of the cortical infarct; (iv) GDNF promotes neurogenesis; (v) GDNF reduces the damage caused by the inflammatory process [27,184,185,186]. Table 4, b and Figure 3 summarize the anti-inflammatory effects of GDNF in certain neurological diseases and the possible associated molecular mechanisms. Importantly, GDNF therapy can be combined with other neurotrophic factors enhancing their neuroprotective properties by anti-apoptotic and antioxidant activity, and reducing excitotoxicity and the M1 microglial phenotype (with pro-inflammatory properties) at the site of damage [181].

### 5.3. Anti-Inflammatory Effects of GDNF Therapy on Systemic Disease Models

GDNF anti-inflammatory effects are not exclusive to the nervous system, but GDNF also acts on systemic inflammation. Peripheral GDNF comes from several cellular sources, such as enteric glial cells, plays many physiological roles, and displays beneficial actions on pathological processes in peripheral tissues (Table 4, c). It has been proposed that GDNF could have the following therapeutic effects in systemic pathological conditions [180,182,183,187,188]: (a) cell survival and restoration of the epithelial barrier reducing its permeability; (b) immunoregulation through the decrease in the expression of the transcription factor NF-κB; (c) reduction of oxidative stress; (d) attenuation of programmed cell death, including apoptosis and autophagy; and (e) decrease in the inflammatory response.

The latter anti-inflammatory property discussed widely in this review has been associated with the decrease in the level of several pro-inflammatory molecules (mainly IL-1β and TNF-α) and infiltration of immune cells [182,183]. Furthermore, GDNF can promote the phenotypic transformation of M1 macrophages to the M2 repair phenotype, modifying the expression of TNF-α and iNOS and favoring the expression of anti-inflammatory molecules such as IL-10 and IL-4, cyclooxygenase 2 (COX- 2), and TGF-β1 [182]. Therefore, it could be assumed that this transformation from M1 to M2 could also occur in microglia cells and even in astrocytes, promoting anti-inflammatory and neurogenerative effects after GDNF therapy in PD.

## 6. Proof of Principle of GDNF in Clinical Trials for Parkinson’s Disease

In preclinical models, GDNF has been shown to protect and restore mature DA neurons in rodent (Table 2, a) and non-human primate (Table 2, b) PD models. Generally, initial outcomes of clinical trials indicate that GDNF therapy could be of value in PD when the therapeutic agent was injected into the putamen rather than the cerebral ventricles [135,189,190].

However, subsequent double-blind placebo-controlled trials, the most recent report being in 2019, concluded that treatment with GDNF was ineffective for PD. As a result, there has been uncertainty about whether GDNF (and related GDNF family neurotrophic factors) could serve to treat PD [35]. Many studies explored the efficacy of direct infusion of GDNF into the putamen, and while open-label studies were promising, a non-randomized study did not meet its primary endpoint [135,189]. Limitations intrinsic to direct putaminal delivery of GDNF or the vector containing the coding sequence for GDNF could be potential reasons for failure. Improved vectors for gene delivery of GDNF are considered as a potentially efficient alternative.

Table 5 summarizes current GDNF clinical trials for PD according to the clinical trials website (ClinicalTrials.gov, accessed on 29 August 2021) and publications available in the literature. Likewise, the table content emphasizes the clinical phase, the characteristics of the study, and the outcome measures or/and preliminary results. The last aspect is associated with adverse effects, safety, tolerability, analysis of laboratory studies (serum and cerebrospinal fluid; CSF), and cabinet. In most cases, only aspects related to clinical and imaging improvement were considered, and no markers were associated with neuroinflammation and/or α-syn, which could result in a critical bias when considering the efficacy of treatment.

Although the strong evidence for the neuroprotective and neurodegenerative effects for GDNF was demonstrated in experimental animals and in vitro, neurorestorative effects were not observed when treating PD patients. As discussed above, we believe that the therapeutic mechanism of action of GDNF has not been sufficiently defined. Alternatively, the degenerating PD brain may be resistant to the neuroprotective potential of this neurotrophic factor. It has been hypothesized that the failure of GDNF in clinical trials may be due to the disruption in the GDNF/RET/Nurr1 signaling pathway [172]. Although this mechanism is supported by findings in experimental animals [172], others refute such a hypothesis [19]. These controversial results suggest that there are other tactical variables not inherent to the disease or resistance to GDNF that should also be taken into account in the treatment of PD patients; for instance, (i) optimal timing of drug administration during the natural course of the disease; (ii) route of administration; (iii) anatomical site of administration; (iv) administration period; (v) type of associated therapies in addition to GDNF treatment, and (vi) delivery system efficiency.

## 7. Conclusions and Perspectives

Despite promising results in preclinical assays, the clinical trials have so far shown disappointing results. It remains to be determined whether the lack of success in PD clinical trials reflects a lack of biological effect of the intervention, limitations of the methods of administration used (e.g., insufficient coverage of the putamen area, right dosage, effective delivery approaches), or an inherent inefficacy of the treatment at an advanced stage of the disease. In the latter case, the possibility of administering GDNF in presymptomatic or early stages of PD should be considered.

The use of GDNF in PD treatment has been mainly supported by preclinical studies in 6-OHDA and MPTP models that neither generate pathological α-syn aggregates, nor chronic neuroinflammation with the participation of neurotoxic reactive astrocytes A1. Therefore, previous GDNF therapy schemes in clinical trials were based on unsuitable preclinical models. However, some preclinical studies have shown that GDNF decreases pathological α-syn and neuroinflammation in transgenic α-syn animal models through mechanisms that remain unclear. Therefore, anti-α-syn aggregation and anti-inflammatory activity are likely effects of GDNF. As α-syn aggregates and neuroinflammation play an important role in the development of PD, we propose that GDNF could counteract these two pathological marks and help to delay or stop the progression of the disease.

Taking into account these results in animal models and Braak’s hypothesis of the prion-like spreading of α-syn, we suggest that GDNF treatment should be applied in the early stages of PD and at neuroanatomical sites where α-synucleinopathy begins. In future studies, it is important to use animal models that develop the four pathognomonic hallmarks of PD to elucidate the therapeutic mechanisms of GDNF. In this regard, the BSSG model seems suitable for evaluating new therapeutic approaches of GDNF in presymptomatic or symptomatic stages of the disease.

Finally, it is important that the clinical trials evaluate biomarkers of α-synucleinopathy and neuroinflammation in serum, CSF, and neuroimaging studies of PD patients to determine the effectiveness of GDNF therapy. The analysis and understanding of these markers will be useful to decide changes in dose and duration with GDNF-based therapy, as well as to determine both the ideal time to start and the endpoint of treatment. Applying these new considerations to GDNF therapy in future clinical trials could lead to favorable results.

## Figures and Tables

**Figure 1 ijms-22-11702-f001:**
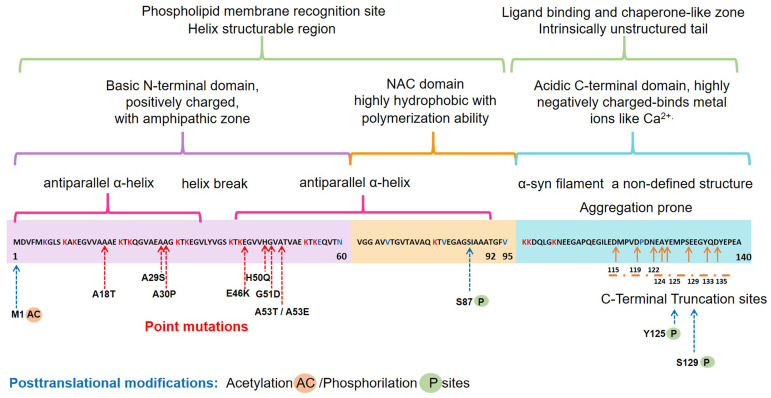
Schematic representation of the native α-syn monomeric structure, highlighting features linked to its biochemical function and dysfunction. Abbreviation: NAC, non-amyloid β-component.

**Figure 2 ijms-22-11702-f002:**
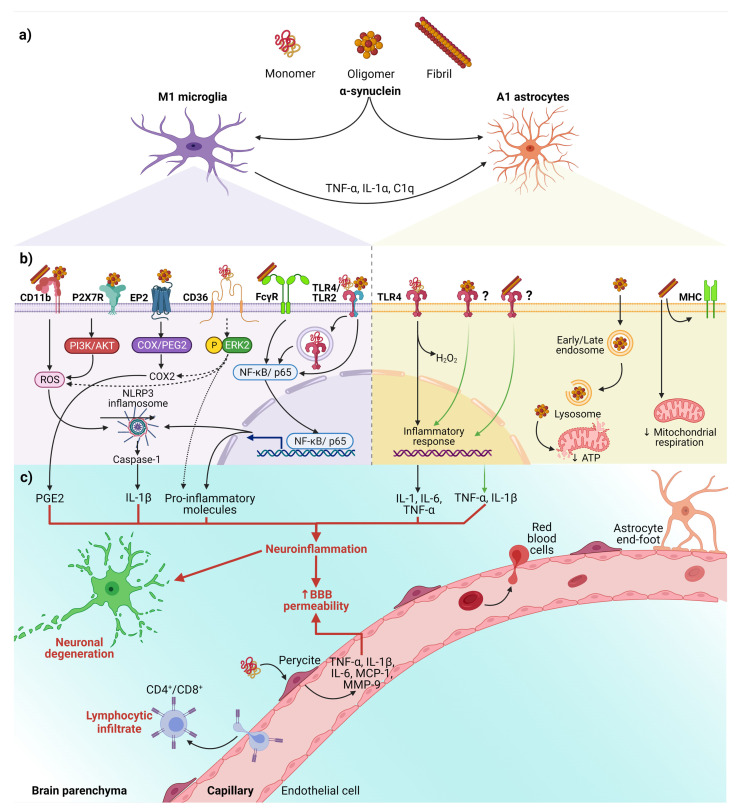
The interaction between α-syn and neuroinflammation in PD. (**a**) Activation of glial cells by pathological α-syn aggregates; (**b**) Signaling pathways in microglia and astrocytes triggered by interaction with the different aggregation patterns of α-syn; (**c**) Neuronal and BBB dysfunction triggered by the neuroinflammatory environment. Dash lines indicate the signaling pathway activated by CD36; Question marks and faded green lines indicate the proposed mechanism for oligomeric and fibrillar α-syn interaction with astrocytes, and thick red lines indicate the proinflammatory molecules that lead to dysfunction of the BBB and neural degeneration. Abbreviations: AKT, Protein kinase B; BBB, Blood-brain barrier; C1q, complement component 1q; CD, cluster of differentiation; COX, cyclooxygenase; EP2, E prostanoid receptor 2; ERK2, extracellular signal-regulated kinase 2; FcγR, the gamma chain subunit of Fc receptor; H_2_O_2_, hydrogen peroxide; IL, interleukin; MCP-1, Monocyte Chemoattractant Protein-1; MHC, Major Histocompatibility Complex; MMP-9, Matrix metallopeptidase 9; NF-κB, nuclear factor kappa-light-chain-enhancer of activated B cells; NLRP3, NLR family pyrin domain containing 3; P2X7R, P2X7 receptor; PGE2, Prostaglandin E2; PI3K, phosphoinositide 3-kinase; ROS, reactive oxygen species; TLR, Toll-like receptors; TNF-α, tumor necrosis factor-alpha. Created with BioRender.com.

**Figure 3 ijms-22-11702-f003:**
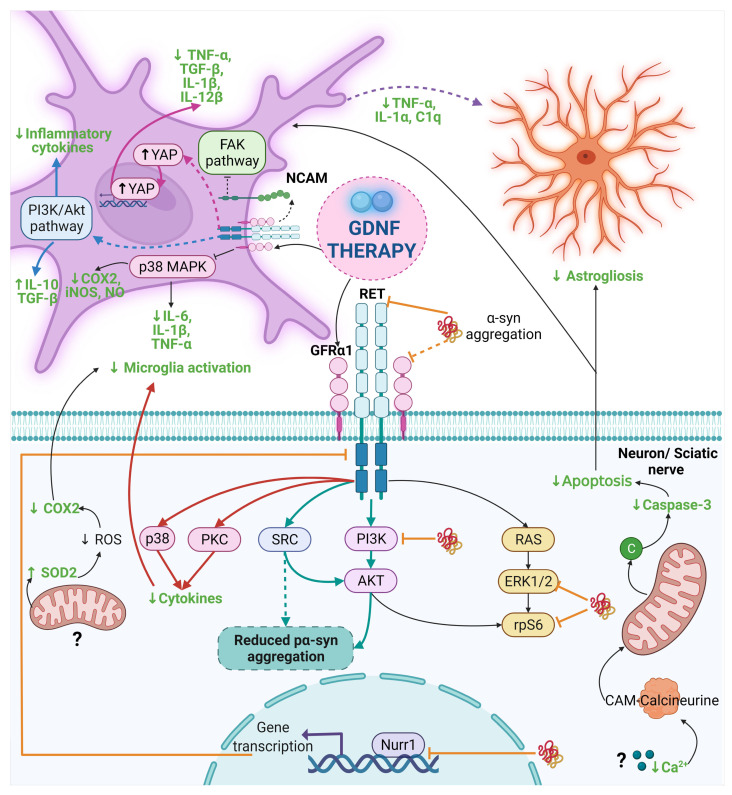
The anti-α-syn and anti-inflammatory effect of GDNF in PD and other diseases. Dash lines indicate the proposed mechanism by other authors and dash thick purple lines are the proposed mechanism in this review. The mechanism of the anti-α-syn and anti-inflammatory effect of GDNF points out with thick light blue lines and black lines in Parkinson’s disease, respectively. Thick orange lines indicate the interruption in the GDNF signaling caused by α-syn. The mechanism for the anti-inflammatory effect of GDNF is highlighted with thick pink lines, thick blue lines, and thick red lines for Alzheimer’s disease, neurological diseases, and neuropathic pain, respectively. The final anti-inflammatory effect of GDNF is in a bold green letter. Abbreviations: α-syn, α-synuclein; C1q, complement component 1q; CAM, Calmodulin; COX, cyclooxygenase; FAK, Focal adhesion kinase; GDNF, Glial cell-derived neurotrophic factor; GFRα1, GDNF family receptor α1; IL, interleukin; iNOS, Inducible nitric oxide synthase; NCAM, neural cell adhesion molecule; NO, Nitric oxide; PI3K, phosphoinositide 3-kinase; PKC, Protein kinase C; RET, Receptor tyrosine kinase; ROS, reactive oxygen species; SOD2, Superoxide dismutase-2; SRC; Proto-oncogene tyrosine-protein kinase Src; TGF-β, Transforming growth factor-beta; TNF-α, tumor necrosis factor-alpha; YAP, Yes-associated protein. Created with BioRender.com.

**Table 1 ijms-22-11702-t001:** Neuroinflammatory and neurotoxic effects triggered by pathological α-syn interaction with glial cells.

α-Syn Aggregation Pattern	Glial Receptor/Mechanism	Signaling Pathway	Neuroinflammatory/Neurotoxic Effects	Ref.
**Fibrils**	NLRP3 ^a^	α-syn acts as DAMP and activates the NLRP3 inflammasome.	Synthesis and release of IL-1β and cleaved caspase-1 that triggers pyroptosis *.	[15,83,84,85,86,87]
**Monomers, oligomers, and fibrils**	TLRs ^a,b,^†	TLRs sense DAMPs, including α-syn, leading to nuclear translocation of NF-κB.	Release of pro-inflammatory cytokines (TNF-α and IL-6). Dual effect on the astrocyte: secretion of pro-inflammatory and/or neuroprotective factors.	[12,15,16,17,88,89,90,91]
**Fibrils**	FcγR ^a^	Internalization in phagosomes and nuclear translocation of NF-kB p65.	Clearance of α-syn, triggering the release of pro-inflammatory molecules and neurodegeneration.	[15,92,93]
**Oligomers and fibrils**	CD11b ^a^	NOX2 activation through a RhoA-dependent pathway.	NOX2 activation mediates the chemoattractant ability of α-syn. Induction of superoxide production.	[15,79,94]
**Oligomers**	EP2 ^a^	The cyclooxygenase/prostaglandin E2 (COX/PGE2) pathway.	Activation of PHOX NADPH and increase in prostaglandin E2 levels, leading to neuronal toxicity.	[15,95,96,97]
**Monomers**	CD36 ^a^	Phosphorylation of ERK2, a downstream kinase activated by CD36 ligation.	Neuronal death through the release of TNF-α and ROS and up-regulation of COX2, NOX2, and iNOS.	[15,72]
**Oligomers**	P2X7R ^a^	Activation of the PI3K/AKT pathway.	Increase of oxidative stress by p47phox translocation and PHOX activation.	[15,98]
**Fibrils**	MHC ^b^	Changes in the expression of *HLA* genes encoding MHC class I and II proteins.	Impairment of ATP-generating mitochondrial respiration.	[99]
**Oligomers**	Endocytosis ^b^	Dysfunction in mitochondrial dynamics.	Neuronal death is mediated by cytokines release.	[100]

Abbreviations: AKT, Protein kinase B; ATP, Adenosine triphosphate; CD, cluster of differentiation; COX, cyclooxygenase; DAMP, damage-associated molecular pattern; EP2, E prostanoid receptor 2; ERK2, extracellular signal-regulated kinase 2; FcγR, The gamma chain subunit of Fc receptor; HLA, human leukocyte antigen; IL, interleukin; iNOS, inducible nitric oxide synthase; MHC; major histocompatibility complex; NADPH, nicotine adenine dinucleotide phosphate; NF-κB, nuclear factor kappa-light-chain-enhancer of activated B cells; NLRP3, NLR family pyrin domain containing 3; NOX2, NADPH oxidase 2; P2X7R, P2X7 receptor; PGE2, Prostaglandin E2; PHOX, phagocytic oxidase; PI3K, phosphoinositide 3-kinase; ROS, reactive oxygen species; TLR, Toll-like receptors; TNF-α, tumor necrosis factor-alpha. * An inflammatory type of cell death that causes excessive cell swelling, membrane rupture, and cytokines release. ^a^ Microglia and ^b^ astrocytes. † Oligomers for TLR-2; monomers, oligomers, and fibrils for TLR-4.

**Table 2 ijms-22-11702-t002:** An overview of GDNF neuroregenerative effects on PD preclinical models.

GDNF Therapy and Lesion Models	Neurodegenerative or Neuroprotective Effects of GDNF Therapy	Ref.
**(a) Murine models**		
ICV infusion of rhGDNF or ^125^Iodine-labelled GDNF in unlesioned rats.	Significantly increased striatal and nigral DA levels.	[148,149]
Bilateral ICV GDNF peptide infusion in aged rats with 6-OHDA lesion.	Improved locomotor performance and increased striatal DA turnover.	[150]
Intraventricular infusion of rhGDNF in rats with 6-OHDA lesion.	AIRB prevention. PET analysis showed DA reuptake reduction in the ipsilateral STR. Reduced loss of TH positive neurons in the SNpc and VTA.	[151]
Intranigral GDNF peptide administration in rats with 6-OHDA intrastriatal lesion.	No protection of STR terminals, absence MD recovery, but prevention of cell death in SN.	[152]
Intranigral hGDNF gene transfection in 6-OHDA lesioned rats.	Improved locomotor performance resulting from the regeneration of the nigrostriatal dopaminergic system.	[153]
Intranigral or ICV GDNF peptide administration in rats with 6-OHDA lesion.	The intranigral therapy prevented AIRB and increased TH activity in SN but not in the STR. The ICV therapy transiently reduced AIRB and increased TH only in the ipsilateral SN.	[154]
Intraventricular or intrastriatal rhGDNF infusion via an osmotic minipump over 4 wk in rats with 2 wk-lesion of 6-OHDA.	The ICV therapy successfully blocked the late neuron degeneration in the SN and caused long-lasting relief of MD. Intrastriatal therapy transiently improved MD only during the infusion period, and DA cells’ rescue was less prominent.	[155]
Intrastriatal administration of microspheres of N-glycosylated rhGDNF in rats with 6-OHDA lesion.	Relieved the AIRB and increased the density of TH+ fibers at the striatal level. The therapy proved to be suitable to release biologically active GDNF over up to 5 wk.	[156]
Intrastriatal administration of hGDNF gene-loaded nanoparticles in rats with 6-OHDA lesion.	Promoted survival of grafted fetal DA neurons, increased the survival of TH + cells, and significantly improved motor behavior.	[157,158]
Intrastriatal grafts of genetically modified fibroblasts to produce GDNF in 6-OHDA rats.	Behavioral improvements and, 6 months after grafting, strong GDNF immunoreactivity in the STR. No changes in DA levels and its metabolites neither TH immunoreactivity in the STR.	[159]
Intraestriatal administration of GDNF-loaded microspheres in animals with 2- wk 6-OHDA lesion.	Increased DA striatal terminals and neuroprotection of DA neurons, long-term improvements of behavior until the end of the study (wk 30).	[160]
FUS *-facilitated the delivery of rGDNF-PLs-MBs to the rat brain with 6-OHDA-lesion.	Neuroprotection in effects on TH+ cell number and levels of DA and its metabolites. Also, it prevented the progression of motor-related behavioral abnormalities.	[161]
**(b) Non-human primate models**		
Intraventricular rhGDNF administration in rhesus monkeys and marmosets with MPTP-lesion.	Relief of MD correlates with increased DA levels and its metabolites in the SN, but not in the Pu of rhesus monkeys. Promising improvements were observed in MD in the marmosets.	[162,163,164,165]
Intrastriatal continuous hGDNF administration in rhesus monkeys with MPTP-lesion.	It increased the DA cell number in the SN and fiber density in the CN, Pu, and GP and increased DA and its metabolite levels. Improvement of the PD rating scale.	[129,166,167,168]
Intraputamenal CED delivery of AAV2-GDNF in rhesus monkeys with MPTP-lesion.	Led to GDNF expression in the Pu, anterograde transport to the SN and rescue of DA neurons via retrograde striatonigral transport, and reversion of neuroregeneration.	[169]

Abbreviations: 6-OHDA, 6-hydroxydopamine; AAV2, Adeno-associated virus; AIRB, apomorphine/amphetamine-induced rotational behavior; CED, convection-enhanced delivery; CN, caudate nucleus; DA, Dopamine/Dopaminergic; FUS, Focused ultrasound; GDNF, Glial cell-line derived neurotrophic factor; GP, globus pallidus; hGDNF, human GDNF; ICV, intracerebroventricular; MBs, microbubbles; MD, motor deficits; MPTP, 1-methyl-4-phenyl-1,2,3,6-tetrahydropyridine; PD, Parkinson’s disease; PET, positron emission tomography; GDNF-PLs-MBs, GDNF plasmid loaded in PEGylated liposomes coupled to microbubbles; hGDNF, human GDNF; Pu, putamen; rGDNF, rat GDNF; rhGDNF, recombinant human GDNF; SN, *substantia nigra*; SNpc, *substantia nigra pars compacta*; STR, striatum; TH, tyrosine hydroxylase; VTA, ventral tegmental area; wk, week(s). * FUS-induced MBs have been used for blood-brain barrier non-invasive opening to allow targeted delivery of therapeutics to the brain.

**Table 3 ijms-22-11702-t003:** GDNF therapy against α-syn pathology in preclinical trials of PD.

Experimental Model	Type of Therapy with GDNF	Featured Results	Proposed Mechanism/Concluding Remarks	Ref.
Genetic rat model of PD (human A30P mutant α-synuclein overexpression by Lenti-A30P).	Intranigral transduction of GDNF-LV (200,000 ng of p24/mL), 2 wk before the intranigral injection Lenti-A30P).	GDNF did not prevent the loss of DA neurons and nerve terminals induced by α-syn toxicity, despite promoting the sprouting of DA axons.	The lack of GDNF neuroprotective effects may be caused by α-syn-mediated blockade of the GDNF/GFRα/RET signaling pathway.	[170]
Genetic rat model of PD (human wild-type α-syn overexpression by AAV-α-syn).	Intrastriatal (2 wk before AAV-α-syn) and intranigral (3 wk before AAV-α-syn) injection of GDNF-LV (1 × 10^7^ transduction units/mL) and AAV-GDNF (1 × 10^12^ genome copies/mL).	Both GDNF therapies did not protect nigral neurons and striatal DA innervation against α-syn-induced toxicity. Also, GDNF did not affect the process of α-syn aggregation.	The α-syn-based rat model can cause poor axonal transport, which may interfere with antegrade or retrograde transport of GDNF in the nigrostriatal system.	[171]
Genetic: human WT α-syn overexpression by AAV-α-syn.	Intrastriatal or intranigral injection of rhGDNF (1 μg/3 μL) 2 wk after administration of AAV- α-syn.	GDNF failed to activate the AKT and MAPK pathways in the genetic model. It also reduced the expression of the *RET* and *NR4A2 (Nurr1)* genes.	Therapeutic failure was caused by the toxicity of α-syn that blockades the GDNF/RET/Nurr1 signaling pathway.	[172]
Parkin Q311X(A) transgenic mice.	GDNF-producing macrophage injections (i.v. 2 × 10^6^ cells/100 μL/mouse) every wk for 3 wks.	GDNF reduced the formation of α-syn aggregates. It also restored the impaired locomotor functions.	The result is attributed to the anti-inflammatory subtype of M2 macrophages.	[27]
Primary embryonic midbrain cultures with α-syn PFFs added on culture day 8. Intrastriatal injection of α-syn PFFs in adult C57Bl/6NCrI mice.	pCDH-hSYN-hGDNF LV (MOI ≈ 5, on days 0, 5, and 9). Intrastriatal injection of AAV1-hGDNF (2.14 × 10^12^ vg/m) one wk before α-syn PFFs injection.	In vitro, GDNF overexpression reduced misfolded α-syn and phosphorylated α-syn. In vivo, GDNF overexpression prevents the α-syn aggregates in the SN and then spreads to the brain.	Activation of the GDNF/RET pathway prevents Lewy pathology. The inhibition of the PI3K/AKT signaling increased the phospho α-syn. The inhibition of SRC blocks the effect of GDNF on α-syn accumulation.	[25]

Abbreviations: AAV, adeno-associated virus vectors; AKT, protein kinase B; DA, dopaminergic; GDNF, glial cell line-derived neurotrophic factor; GFRα, GDNF family receptor alpha; i.v, intravenous; LV, lentiviral vector; MAPK, mitogen-activated protein kinase; MOI, multiplicity of infection; NR4A2, Nuclear Receptor Subfamily 4 Group A Member 2; Nurr1, Nuclear receptor-related 1 protein; PD, Parkinson’s disease; PFFs, Preformed Fibrils; PI3K, phosphoinositide 3-kinase; RET, receptor tyrosine-protein kinase; SN, *substantia nigra*; SRC, non-receptor protein tyrosine kinase; α-syn, α-synuclein; WT, wild type; wk, week(s).

**Table 4 ijms-22-11702-t004:** Anti-inflammatory effect of GDNF in preclinical trials of Parkinson’s disease and other neurological diseases.

Experimental Model/*GDNF Therapeutic Approach*	GDNF Relevant Effects	Proposed Mechanisms	Ref.
**(a) Neurodegenerative disease**			
Parkinson’s disease			
Midbrain microglial cultures activated by Zymosan A/*Therapy with* *astrocyte-derived GDNF.*	GDNF decreases microglial activation in a model of neuroinflammation in vitro.	Activation of the GFRα1-RET complex and inhibition of the FAK pathway.	[33]
12-month-old GDNF^+/-^ knockdown mice/*Bilateral intrastriatal injection of GDNF (10 μg per hemisphere).*	Attenuation of motor impairments and nigrostriatal dopamine levels.	Reduction of the COX-2 expression and increase in SOD-2 levels in the SN suggests reduced microglial activation.	[32]
Rat microglial culture activated by LPS/*Therapy with rHGDNF (50 ng/mL, before LPS administration).*	Prevention of NO synthesis and iNOS COX-2, IL-6, IL-1β, and TNF-α expression.	Inhibition of microglial activation by reducing phosphorylation of p38 *.	[31]
6-OHDA-induced PD rat model/*Intrastriatal* *GDNF* *released by* *LCM (1.5 mg/kg, 3d after 6-OHDA injection).*	Reduction of caspase-3 and TNF-α levels and activation of microglia.	Reduced microglial activation by lowering MMP-9 and MHC II expression, which consequently retains DA.	[30]
Intracutaneous 6-OHDA injection in mice/*GDNF transfected macrophages (i.v. 1 × 10^6^ cells/100* *μ**L, 2d after 6-OHDA).*	It decreased activated microglia in SNpc.	The macrophages migrate to inflammation sites and provide neuroprotection modulating glial cells activation.	[27]
Intraperitoneal MPTP injection in mouse/*IN GDNF delivery by peptide-conjugated lipid nanocarriers (**2.5 μg, 2.5 μL per nostril, 4 sessions**).*	It decreased activated microglia.	It targets microglia activation, modulating the neuroinflammation.	[28]
Transgenic Parkin Q311X(A) mice (4-month-old)/*Administration of GDNF-macrophages (i.v. 2 × 10^6^ cells/100* *μ**L/mouse, once a wk for 3wk).*	Disminution of microgliosis and astrogliosis.	Anti-neuroinflammatory effect by modulating glial activation.	[27]
Intraperitoneal MPTP injection in mouse twice a wk for 3wk/*GDNF gene delivery* via *the UTMD system* *(**3.6 × 10^8^ MBs/mL, twice a wk for 3wk, 1d after MPTP**).*	It decreased apoptosis and astrogliosis.	Reduced expression of calcium ions and the apoptotic protein caspase 3.	[26]
Alzheimer’s disease			
Murine microglial cell line BV2 stimulated with Aβ-protein/*GDNF (100 and 500 ng/mL, after* *A**β**-protein administration**).*	Decreased levels of TNF-α, TGF-β, IL-1β, and IL-12β, in a dose-dependent manner.	Inhibition of microglial activation by reducing YAP phosphorylation (Hippo/YAP pathway).	[175]
Amyotrophic lateral sclerosis			
Transgenic SOD1^G93A^ rat (90d old)/*i.m. grafts of hMSC-GDNF (150,000 cells in 50 μL, at 24h, 1 and 2wk).*	Reduced inflammation markers and promoted TSC survival. Delayed onset of ALS symptom	Delayed ALS symptom onset by preserving NMJ integrity.	[176]
Multiple sclerosis			
CEAE animal model for multiple sclerosis: Transplantation of GDNF/*NSCs in each lateral ventricle (5 × 10^5^ in 10* *μ**L, 10d after the CEAE induction).*	Reduced inflammatory infiltrates in the STR and the astrocyte differentiation of NSCs.	Possible activation of myeloid dendritic cells and subsequent restriction T cell expansions.	[177]
**(b) Other neurological diseases**			
Murine microglial cell line BV2 with ADSCs-*GDNF (500 ng/L).*	Inhibition of the M1 phenotype and promoting the M2 phenotype.	Microglial activation and polarization via PI3K/AKT signaling pathway.	[179]
Neuropathic pain model by CCI in rats/*rAV-GDNF* *[2 × 10^9^ pfu/100 mL PBS] injection in the triceps brachii muscle.*	Down-regulated protein expression of MMP9, p38, IL6, IL1β, and iNOS.	Inhibition of microglial activation and cytokine production via p38 and PKC signaling.	[180]
FCI rat animal model and neuronal culture GOSD/*NCM with TGFβ1, GDNF, and NT-3 (1 ng/mL for each brain cell in plates; 6.4 × 10^6^/plate) and incubated under GOSD conditions (2 h).*	Reduction of the brain infarction and motor deficit.	Anti-apoptotic, anti-oxidant, and potentially anti-glutamate activities.	[181]
**(c) Systemic diseases**			
UUO mice model/*GDNF-AMSCs (i.v. 150 μl cell of suspension containing 5 × 10^5^ cells* via *the tail vein*).	Activation of M2 macrophages and reduced renal fibrosis by suppressing inflammation.	Down-regulated TNF-α and iNOS; upregulated IL-4 and IL10.	[182]
Experimental colitis mouse model induced by DSS/*rAV-GDNF (* *intracolonically 1 × 10^10^ pfu/mouse)*	Reduction in enhanced permeability by restoring epithelial barrier function.	Inhibition of TNF-α, IL-1β, MPO, caspase-3, and NF-kB. Increase in ZO-1 and AKT.	[183]

Abbreviations: 6-OHDA, 6-hidroxydopamine; ADSCs, adipose-derived stem cells; AKT, serine-threonine protein kinase; ALS, Amyotrophic lateral sclerosis; AMSCs, adipose-derived mesenchymal stem cells; Aβ, Amyloid beta; CCI, chronic constriction injury; CEAE, Chronic Experimental Allergic Encephalomyelitis; COX-2, Cyclooxygenase-2; DA, dopamine; DSS, dextran sulphate sodium; FAK, Focal adhesion kinase pathway; FCI, Focal cerebral ischemia; GDNF, Glial cell line-derived neurotrophic factor; GFRα, GDNF family receptor alpha; GOSD, Glucose-Oxygen-Serum-Deprivation; hMSC, Human mesenchymal stem cell; i.m., intramuscular; i.p., intraperitoneal; i.v., intravenous; IL, Interleukin; IN, intranasal; iNOS, inducible nitric oxide synthase; LCM, lipid-coated microbubbles; LPS, Lipopolysaccharide;MBs, microbubbles; MHC-II, major histocompatibility complex; MMP, Matrix metalloproteinase; MPO, Myeloperoxidase; MPTP, 1-methyl-4-phenyl-1,2,3,6-tetrahydropyridine; NCM, neuron-derived conditioned medium; NF-κB, nuclear factor kappa-light-chain-enhancer of activated B cells; NMJ, neuromuscular junction; NO, nitric oxide; NSCs, Neural Stem Cells; NT-3, neurotrophin-3; p38, Mitogen activated protein kinase p38; PBS, phosphate buffer saline; PD, Parkinson’s disease; pfu, plaque-forming units; PI3K, phosphoinositide 3-kinase; PKC, protein kinase C; rAV, recombinant adenovirus; RET, Receptor tyrosine-protein kinase; rhGDNF, recombinat human GDNF; SN, *Substantia nigra*; SNpc, *Substantia nigra pars compacta*; SOD, superoxide dismutase; STR, striatum; TGF-β, Transforming growth factor-β; TNF-α, Tumor necrosis factor-α; TSC, Terminal Schwann cell; UTMD, ultrasound-targeted microbubble destruction; UUO, unilateral ureteral obstruction; YAP, yes-associated protein; ZO-1, Zonula occludens-1. * p38, mitogen-activated protein kinases, p38/MAPK pathway.

**Table 5 ijms-22-11702-t005:** GDNF current clinical trials for Parkinson disease, based on NIH ClinicalTrials.gov (accessed on 30 August 2021).

CT Identifier. Clinical Phase and Status.	Characteristics of Subjects Recruited	Aims of the Study	Outcome Measures/Preliminary Results	Ref.
**NCT04167540** Clinical Phase: I Clinical status: Active, not recruiting.	25 individuals, males, and females at least 18 years of age (recent/long-standing diagnosis of PD) who are not well controlled by medications (follow up for 5 years).	Evaluate the safety and clinical effect of AAV2-GDNF delivered to the Pu.	Access to disease improvement: MRI, serum and CSF tests, movement and behavioral tests. No preliminary results	[191,192]
**NCT01621581** Clinical Phase: I Clinical status: Completed.	25 individuals, males, and females at least 18 years of age diagnosed with idiopathic and advanced PD (follow up for 5 years).	Evaluate the safety and effectiveness of AAV2-GDNF gene transfer to the Pu via CED.	Good drug tolerability based on clinical and neuroimaging tests. The PET showed increased F-DOPA uptake in the infused areas at 6 and 18 months in 10/13 and 12/13 patients, respectively.	[135,192,193]
**NCT03652363** Clinical Phase: II Clinical status: Completed.	42 individuals, females and males between 35 to 75 years diagnosed with idiopathic PD (followup for 5 years).	Analyze the safety and efficacy of intermittent bilateral intraputamenal GDNF via CED.	Periodic evaluation with specific clinical and blood tests or an MRI scan. Disappointing final results, with no significant clinical improvement between GDNF and placebo group.	[135,192,194]
**NCT00006488** Clinical Phase: I Clinical status: Completed.	Females and males between 18 and 75 years of age were diagnosed with idiopathic PD. The number of individuals is not specified.	Determine the benefits and TEAE of continuous infusion of r-metHuGDNF in Pu.	Subjects were evaluated using clinical neurological tests, computerized gait assessment, and neurological imaging. The results of this study were not specified.	[192,195]

Abbreviations: AAV, Adeno-associated virus; CED, Convection Enhanced Delivery; CSF, cerebrospinal fluid; F-DOPA, fluoro-3,4-dihydroxyphenylalanine; GDNF, Glial cell-derived neurotrophic factor; IC, intracerebral; MRI, Magnetic resonance imaging; PD, Parkinson’s disease; PET, positron emission tomography; Pu, Putamen; r-metHuGDNF, recombinant-methionyl human Glial Cell line-derived Neurotrophic Factor; TEAE, Treatment-Emergent Adverse Events.

## Data Availability

Not applicable.

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
