# Peer review of "Mechanistic Insight from Preclinical Models of Parkinson’s Disease Could Help Redirect Clinical Trial Efforts in GDNF Therapy"

_ijms, 2021, doi:10.3390/ijms222111702_

Round 1

Reviewer 1 Report

Well written review, with up to date knowledge on GDNF-based therapy of PD patients

the review produce a positive impression on state of the art in field of discovery of novel therapeutic strategies to treat PD. It is a big peace of work, well structured, logical with good English it is easy to follow.

Author Response

We appreciate the reviewer's comments and dedication in reviewing our manuscript.

Reviewer 2 Report

The manuscript by Delgado-Minjares et al. represents an extensively review on Parkinson's disease from both a pathogenetic and the potential new GDNF-related therapeutic perspectives point of view. However, the title of the manscript is a little bit misleading since, the reader expectations are greatly unsatisfied by the results of clinical trial and more importantly by the authors' conclusions. In my opinion, the authors should discuss in a deeper way the reasons related to the failure of clinical trials considering one of the most important aspect related to GDNF delivery in the patients, and speculate on how is  possible to overcome this limitation. Otherwise, the manuscript lacks in its novelty. Therefore, the title needs to be modified or an answer to the question posed in the title has to be provided

Moreover, the manuscript lacks in important references, considered milestones in PD, mainly related to alpha-synuclein and its pathogenetic role in PD as well as in PD animal models.

Author Response

The manuscript by Delgado-Minjares et al. represents an extensively review on Parkinson's disease from both a pathogenetic and the potential new GDNF-related therapeutic perspectives point of view.

  • We appreciate the reviewer's comments and dedication in reviewing and improving our manuscript.

Q1. However, the title of the manscript is a little bit misleading since, the reader expectations are greatly unsatisfied by the results of clinical trial and more importantly by the authors' conclusions. In my opinion, the authors should discuss in a deeper way the reasons related to the failure of clinical trials considering one of the most important aspect related to GDNF delivery in the patients, and speculate on how is possible to overcome this limitation. Otherwise, the manuscript lacks in its novelty. Therefore, the title needs to be modified or an answer to the question posed in the title has to be provided

A1. We value the reviewer's observation and suggestion. Therefore, we have changed the title of our manuscript from “New GNDF therapeutic approaches against Parkinson’s disease: A dark alley or the light at the end of the road?” to “Mechanistic insight from preclinical models of Parkinson's disease could help redirect clinical trial efforts in GDNF therapy”. Emphasizing the importance of understanding the anti-α-synuclein and anti-inflammatory mechanisms of GDNF-based therapy in preclinical trials and how these could affect clinical trial design. Based on the reviewer's comments we have also modified the conclusion section. (please see the corresponding section).

Q2. Moreover, the manuscript lacks in important references, considered milestones in PD, mainly related to alpha-synuclein and its pathogenetic role in PD as well as in PD animal models.

A2. Taking into account the reviewer's observation, we have supplemented with some relevant references related to α-syn and its pathogenetic role in PD (please see Lines 71, 79, 158, 166, 170, and 177). Concerning the references of PD animal models, we consider those cited in Tables 2, 3, and 4 are sufficient.

Reviewer 3 Report

Review of a manuscript “New GNDF therapeutic approaches against Parkinson’s disease: A dark alley or the light at the end of the road?” by Karen M. Delgado-Minjares and coauthors.

Parkinson's disease is the second most prevalent neurodegenerative disorder for which there is not treatment affecting the main course of the diseases. Moreover, there is no reliable biomarkers that would allow to identify early steps of the disorder and begin treatment. Therefore, basic research of molecular mechanism of this disease are currently needed.  The authors explored the inconsistency between preclinical and clinical outcomes, reviewed the mechanisms of a-synuclein aggregation and assessed the neurorestorative properties of glial cell-derived neurotrophic factor (GDNF). The authors summarize the newest data about the role of pathological α-synuclein aggregates in the neuroinflammatory process. The topic of the review manuscript is very important, the authors discussed new data about GDNF anti-inflammatory role and its effect on a-synuclein.

The following corrections should be made:

Introduction

Lines 69-70. After the sentence: ”Experimental evidence suggests that the neurotoxicity of α-syn comes from its transformation into insoluble aggregates, either oligomers or fibrils” the authors should cite the following article “Intracellular dynamics of synucleins: Here, there and everywhere. International Review of Cell Molecular Biology, 2015, 320, 103-169.”

Lines 77-78. In the sentence “Interestingly, extracellular α-syn aggregates can directly stimulate microglia and astrocytes to produce pro-inflammatory cytokines and initiate neuroinflammation [11,13-15] after “extracellular α-syn aggregates” the authors should add the citation of a relevant article “Cell Responses to Extracellular α-Synuclein” published in in Molecules. 2019; 24(2):305.

  1. Glial cell-derived neurotrophic factor in Parkinson's disease

In Table 2 since the authors present under a) Murine models, it is not clear whether they mean to present “Non-human primate models” under b)?

Table 5. GDNF current clinical trials for Parkinson disease, based on NIH ClinicalTrials.gov.

Table 5 is overloaded with details and so hard to read. The authors should truncate it and make it more reader friendly.

  1. Conclusion and perspectives

Lines 570-571. “However, the GDNF mechanism of action remains partly identified because crucial elements of the disease have not been considered, such as the development of α-synucleinopathy and the neuroinflammatory process.”

The sense of this sentence remains unclear.  What the authors mean by “development of α-synucleinopathy” and how it is connected with GDNF mechanism? The authors should rewrite this sentence in a more clear way.  

Line 588 “complete study should evaluate the GDNF neuroregenerative effect and its role in preventing α-syn aggregation and neuroinflammation.”

What exactly the authors mean by “GDNF neuroregenerative effect” and how the authors plan to use it?

Author Response

Parkinson's disease is the second most prevalent neurodegenerative disorder for which there is not treatment affecting the main course of the diseases. Moreover, there is no reliable biomarkers that would allow to identify early steps of the disorder and begin treatment. Therefore, basic research of molecular mechanism of this disease are currently needed.  The authors explored the inconsistency between preclinical and clinical outcomes, reviewed the mechanisms of a-synuclein aggregation and assessed the neurorestorative properties of glial cell-derived neurotrophic factor (GDNF). The authors summarize the newest data about the role of pathological α-synuclein aggregates in the neuroinflammatory process. The topic of the review manuscript is very important, the authors discussed new data about GDNF anti-inflammatory role and its effect on a-synuclein.

  • We appreciate the reviewer's comments and dedication in reviewing and improving our manuscript.

The following corrections should be made:

Introduction

Q1. Lines 69-70. After the sentence:” Experimental evidence suggests that the neurotoxicity of α-syn comes from its transformation into insoluble aggregates, either oligomers or fibrils” the authors should cite the following article “Intracellular dynamics of synucleins: Here, there and everywhere. International Review of Cell Molecular Biology, 2015, 320, 103-169.”

A1. We appreciate the reviewer's suggestion and have added the corresponding reference    (please see Line 71).

Q2. Lines 77-78. In the sentence “Interestingly, extracellular α-syn aggregates can directly stimulate microglia and astrocytes to produce pro-inflammatory cytokines and initiate neuroinflammation [11,13-15] after “extracellular α-syn aggregates” the authors should add the citation of a relevant article “Cell Responses to Extracellular α-Synuclein” published in Molecules. 2019; 24(2):305.

A2.  We have read the reference suggested by the reviewer and it correctly integrates and complements the described phrase (please see line 79).

Q3. 4. Glial cell-derived neurotrophic factor in Parkinson's disease. In Table 2 since the authors present under a) Murine models, it is not clear whether they mean to present “Non-human primate models” under b)?

A.3 The reviewer's observation is correct and in Table 2 was added the subsection b) corresponding to the non-human primate models. In addition, the text was adapted as follows: “In preclinical models, GDNF has been shown to protect and restore mature DA neurons in rodent (Table 2a) and non-human primate (Table 2b) PD models (please see Lines 550 and 551)”; “Rodents (Table 2a) and non-human primates (table 2b) with unilateral or bilateral lesion of the nigrostriatal pathway using specific neurotoxins to model PD have been used to show the preventive and restorative effects of GDNF….” (please see Lines 353 to 355)

Q.4. Table 5. GDNF current clinical trials for Parkinson disease, based on NIH ClinicalTrials.gov. Table 5 is overloaded with details and so hard to read. The authors should truncate it and make it more reader friendly.

A4. We welcome the reviewer's suggestion. We have made the following modifications to Table 5 to make it more reader-friendly (please see Table 5):

  1. i) The general content of the table was summarized.
  2. ii) In the first column, the dates corresponding to the starting and end of the study were eliminated. Besides, the clinical status of the study was added in this column.

iii) The second column was divided in two. Now the second column corresponds to the “Characteristics of subjects recruited” and the third column to the “Aims of the study”.

Q5. 7. Conclusion and perspectives. Lines 570-571. “However, the GDNF mechanism of action remains partly identified because crucial elements of the disease have not been considered, such as the development of α-synucleinopathy and the neuroinflammatory process.” The sense of this sentence remains unclear.  What the authors mean by “development of α-synucleinopathy” and how it is connected with GDNF mechanism? The authors should rewrite this sentence in a more clear way.  

Q6. Line 588 “complete study should evaluate the GDNF neuroregenerative effect and its role in preventing α-syn aggregation and neuroinflammation.” What exactly the authors mean by “GDNF neuroregenerative effect” and how the authors plan to use it?

A5 and A6. We appreciate the reviewer's observations and comments and we have rewritten the conclusion section. Emphasizing the possible causes of failure with GDNF-based therapy in clinical studies. Likewise, we suggest that in these clinical trials based on GDNF therapy, the effects on α-synucleinopathy and neuroinflammation should be taken into account and thus establish optimal treatment designs (please see conclusion and perspectives section).

Reviewer 4 Report

The manuscript of Delgado-Minjares and co-workers concerning the use of GDNF as a possible therapeutic approach in Parkinson’s disease is an interesting and thorough work of significance, since there is still no effective therapy against Parkinson’s disease which can arrest/revert the progression of the disease.

However, there are some minor questions/comments concerning the manuscript:

In Figure 1, it would be worth showing the A18T and the A29S familial mutations as well.

Is alpha-synuclein expressed in microglia and astrocytes under physiological conditions or pathologically? Is there any evidence that the transmission of distinct alpha-synuclein forms (such as Ser129 phosphorylated) affect the cells differently?

The authors focus on microglia and astrocytes, but do not mention oligodendrocytes. In the case of multiple system atrophy, which also belongs to synucleinopathies, glial cytoplasmic inclusions comprising alpha-synuclein can be observed in oligodendrocytes.

Author Response

The manuscript of Delgado-Minjares and co-workers concerning the use of GDNF as a possible therapeutic approach in Parkinson’s disease is an interesting and thorough work of significance, since there is still no effective therapy against Parkinson’s disease which can arrest/revert the progression of the disease.

  • We appreciate the reviewer's comments and dedication in reviewing and improving our manuscript.

However, there are some minor questions/comments concerning the manuscript:

Q1. In Figure 1, it would be worth showing the A18T and the A29S familial mutations as well.

A.1 We consider the reviewer's suggestion very timely and have added the corresponding mutations in Figure 1.

Q2. Is alpha-synuclein expressed in microglia and astrocytes under physiological conditions or pathologically?

A2. We have answered the interesting question posed by the reviewer in section "2. 1. Structure and physiology of α-syn" (please see 115 and 116)

Q3. Is there any evidence that the transmission of distinct alpha-synuclein forms (such as Ser129 phosphorylated) affect the cells differently?

A3. The one asked by the reviewer is a controversial question. However, we have added a phrase related to the reviewer's questioning (please see L156 to 158).

Q4. The authors focus on microglia and astrocytes, but do not mention oligodendrocytes. In the case of multiple system atrophy, which also belongs to synucleinopathies, glial cytoplasmic inclusions comprising alpha-synuclein can be observed in oligodendrocytes.

A4. We appreciate the reviewer's comment and we have added a sentence corresponding to α-synuclein inclusions in oligodendrocytes (please see L164 to 166).
